Latest advancements and prospects in the next-generation of Internet of Things technologies

Amin Farhan 1
Abbasi Rashid rashidd.abbasi@gmail.com 2
Khan Salabat 3
Abid Muhammad Ali 4
Mateen Abdul 5
de la Torre Isabel isator@uva.es 6
Kuc Castilla Angel 7
Garcia Villena Eduardo 7
1 School of Computer Science and Engineering, Yeungnam University , Gyeongsan , Republic of South Korea
2 College of Computer Science and Artificial Intelligence, Wenzhou University , Wenzhou , China
3 School of Computer and Information Engineering, Qilu Institute of Technology , Jinan , China
4 Faculty of Smart Engineering, The University of Agriculture , Dera Ismail Khan , Pakistan
5 Department of Computer Science, Federal Urdu University of Arts, Science and Technology , Islamabad , Pakistan
6 Department of Signal Theory and Communications, University of Valladolid , Valladolid , Spain
7 Universidad Europea del Atlántico , Santander , Spain
Raza Khalid
Electronic publication date: 2024 Oct 25
Publication date: 2024
Volume: 10
Electronic Location ID: e2434
Received 2024 Jul 17; Accepted 2024 Sep 27
Copyright: ©2024 Amin et al.
Copyright year: 2024
Copyright holder: Amin et al.
License: This is an open access article distributed under the terms of the Creative Commons Attribution License, which permits unrestricted use, distribution, reproduction and adaptation in any medium and for any purpose provided that it is properly attributed. For attribution, the original author(s), title, publication source (PeerJ Computer Science) and either DOI or URL of the article must be cited.
License URL: https://creativecommons.org/licenses/by/4.0/

Keywords: Internet of Things, Social Internet of Things, Big data, Social networks

Funding: the European University of the Atlantic, Spain This research is funded by the European University of the Atlantic, Spain. The funders had no role in study design, data collection and analysis, decision to publish, or preparation of the manuscript.

==============================
The Internet of Things (IoT) is a sophisticated network of objects embedded with electronic systems that enable devices to collect and exchange data. IoT is a recent trending leading technology and changing the way we live. However, it has several challenges especially efficiency, architecture, complexity, and network topology. The traditional technologies are not enough to provide support. It is evident from the literature that complex networks are used to study the topology and the structure of a network and are applied to modern technologies. Thus, the capability of powerful computational tools and the existence of theoretical frameworks enable complex networks to derive new approaches in analyzing IoT-based technologies in terms of improving efficiency, architecture, complexity, and topology. In this direction, limited research has been carried out. The integration aspect remains a key challenge. Therefore, in order to fill this gap. Herein, we design a comprehensive literature review. In this research effort, we explore a newly leading emerging technology named the Social Internet of Things (SIoT). It is developed to overcome the challenges in IoT. We discuss the importance and the key applications of SIoT. We first presented a conceptual view along with a recent technological roadmap. The big data play an important role in the modern world. We discuss big data and the 5 Vs along with suitable applications and examples. Then, we highlighted the key concepts in complex networks, scale-free, random networks, and small-world networks. We explored and presented various graph models and metrics aligned with social networks and the most recent trends. The novelty of this research is to propose a synergy of complex networks to the IoT, SIoT, and big data together. We discuss the advantages of integration in detail. We present a detailed discussion on complex networks emerging technologies and cyber-physical systems (CPS). Briefly, our literature review covers the most recent advancements and developments in 10 years. In addition, our critical analysis is based on up-to-date surveys and case studies. Finally, we outline the impact of recent emerging technologies on challenges applications, and solutions for the future. This paper provides a good reference for researchers and readers in the IoT domain.

Introduction

Recently, the Internet of Things (IoT) (Amin, Ahmad & Choi, 2019) attracted the researcher’s community, and as a result, a lot of research has been carried out. IoT refers to the network of physical devices that are embedded with sensors, and software and provide connectivity to the internet. This technology allows the sensors and devices to collect and exchange data with other devices and systems. The Social Internet of Things (SIoT) is an extension of IoT and incorporates social networking concepts into IoT devices. SIoT enables IoT devices to communicate with other devices and establish social relationships. Thus, it results in terms of creating a social network that shares the information in the network. Big data refers to the large and complex data sets that are difficult to process using traditional data processing models. It is characterized by volume, velocity, variety, and veracity. Recently, these modern technologies have been used separately in the development of novel applications.

The motivation for our research

The modern emerging technologies are evolving, rapidly spreading, and have tremendously affected the lives of people and society. To the best of our knowledge, there is limited literature available on the current state of these ecosystems. Technological integration remains a key challenge. This motivates us to write this article.

Problem statment

The efficiency, architecture, complexity, and topology are the main concerns in IoT. The IoT development lies in the high integration of the information, physical, and social space. The implementation of information integration of these three spaces is to mine characteristics and analyze the regular pattern of the relationship between objects in physical space based on the data of IoT. In addition, there is a lack of a relational network consistent with the characteristics of IoT. In general, IoT-based technologies have a large infrastructure and are often comprised of millions of nodes/entities interacting with each other. Hence, the complexity, heterogeneity, and the energy consumption issue arises.

Research questions

Herein, we surveyed leading emerging technologies in the preliminary research phase to identify gaps in IoT technologies. We reviewed various surveys and research papers on the most common in IoT and emerging technologies. The selected key application areas are the SIoT, big data, and complex networks. Our literature survey assisted in contrasting these prominent application domains. Furthermore, we found a variety of surveys that specifically report these recent trends, which served as the impetus for this research. We developed research questions, which are provided below.

RQ1: What other trending emerging technologies are available that are used in smart cities and also constitute an independent ecosystem? This research question aims to identify the trendy and rapidly evolving technologies similar to the IoT.

RQ2: How do the architectural and technical requirements in IoT domains differ from current application domains? This question allowed us to classify the differences in technological requirements and gaps in leading emerging technologies.

RQ3: How can recent emerging technologies be integrated into the modern digital era or blueprint? The integration effect, feasibility, and scope of emerging technologies are the focus of this issue. It also allows for the investigation of technical specifications, mostly at the communication layer, to seamlessly integrate with recent IoT applications.

RQ4: What open research challenges in this area? The majority of IoT communication relies on wireless connectivity. The IoT presented in this article, however, does not use this connectivity. This question aims to clarify the important technological challenges in this era in detail. It is evident from the literature that computational tools and theoretical frameworks enable complex networks to derive new approaches in analyzing IoT in terms of improving architectures, networks, and services. Generally, complex networks refer to networks that exhibit non-trivial structural patterns, such as scale-free degree distributions, community structure, and small-world properties. These networks can be found in social networks, biological networks, and technological networks. This is an interdisciplinary research area and is mainly based on computational computer science, social science, and data science fields. The complex network modeling and analysis is also a new research dimension. The complex network analysis (CNA) software named Cite Space, SCI2, and Gephi are useful for big data analysis. The complex network models for example scale-free and random networks are widely used in the development of and the understanding of social network structure. It is also evident from the literature that IoT network is considered complex networks. Therefore, there is a need to integrate complex networks and emerging. However, limited research has been carried out concerning the integration of complex networks and modern internet-based emerging technologies. Thus, in this research, we presented a synergy between complex networks and emerging technologies. We introduce the basics of complex networks, random networks, and scale-free networks along with suitable examples for the interest of readers. We explored SIoTbig data, and applications along with suitable supporting literature and also discussed the strengths and weaknesses separately.

Difference between our research and others’ research

Our critical analysis is completely different from previous research and surveys. In earlier research surveys partially covered IoT and very few of them covered SIoT and big data. However, very limited/ no research focused on the integration aspect from a social aspect, i.e., SIoT. In contrast, herein, we completely explored and covered complex networks in connection with social networks and SIoT. We critically reviewed various research surveys and studies. Figure 1 presents the number of published articles gathered from a search conducted on the IEEE Explorer and PubMed search engines. The academic publications on complex networks (blue line), the Internet of Things (grey line), the Social Internet of Things (yellow line), and big data (orange line) are shown. In this figure, readers can see that “complex networks” remained the most popular trend over a long time, but “Big Data” led it from 2012 to 2021. It is evident from the figure that complex networks already received significant attention for a long time. However, the IoT, big data, and SIoT received significant attention after 2013 by identifying problems and also creating new opportunities for the research community. In conclusion, we should be focused on the integration of paradigms.

Figure 1 Recent research trends in academic publications.

Aims and objectives of our research

This paper aims to provide a comprehensive literature survey on the recent advancements and developments in IoT-based modern technologies. As the integration is not presented. Thus, we presented the integration of complex networks and emerging technologies. Further, we explore small-world and scale-world networks, especially in connection with SIoT and social networks. We explored and discussed very useful network metrics for complex networks and found them useful for social networks; for instance, path length clustering, and coefficient.

Key contribution of this review

• The novelty of this research is that we propose the integration of leading emerging technologies and highlight key concepts, basic functions, growth, and most recent trends.

• We discuss an overview of complex networks, from the perspectives of scale-free, random, and small-world networks in connection with social networks. We investigated network-related properties for instance; clustering coefficient, network diameter, path length, and their impact on the overall network in connection with social networks.

• We discuss the most recent state of the art along with suitable examples that remained unexplored in the earlier studies.

• Our critical analysis comprised of most recent advances and developments in complex networks and leading emerging technologies. Our analyses cover the most recent state-of-the-art within the last 10 years along with the pros and cons.

• We discuss SIoT which remained unexplored in the earlier surveys.

• Our critical analysis varies from previous research studies since most of the previous researchers have partially or not covered these aspects and relevant technologies and given detailed knowledge about the interlinked aspect has not been discussed explicitly.

• We explore various unique research problems and challenges for instance; energy management, trust, security, and privacy that need to be identified and addressed in the future. The contents enclosed in this survey highlight recent developments in complex networks and emerging technologies, and it provides a solid foundation for future research in this area.

This paper is organized as follows: ‘Complex Networks’ introduces the basics of complex networks, definitions, and key concepts. The leading trending emerging technologies and most recent advancements are discussed in ‘Leading Emerging Technologies and Recent Research’. The synergies between complex networks and IoT-based technologies are discussed in ‘The Synergies between Complex Networks and the Leading Emerging Technologies’. In ‘Discussion, Open Research Challenges, Potential Solutions, and Future Research Trends’, we explore various challenges such as energy management, trust security and privacy, complex data, etc. Finally, ‘Conclusions and Future Works’ concludes the conclusion of this research.

Leading emerging technologies

Figure 2 presents the cycle of emerging technologies. In this figure, we can see that IoT devices as a nature generate a large amount of data. In the second phase, the power of establishing social relationships enables the SIoT to overcome this issue. In the third phase, big data is a leading technology used to handle large amounts of data generated by the IoT or SIoT. In the fourth phase, the complex networks have various unique features. Herein, one of the promising features is to consider IoT as a complex network. This feature allows the complex networks to solve the existing IoT problem in architecture, platform, tools, and infrastructure. In addition, it can solve the issue of big data. The social network analysis (SNA) and the scale-free structure enable solving issues in the SIoT technology. In this section, we provide in-depth knowledge of the most recent literature on emerging technologies and also present a discussion on different approaches in this area. The emerging technologies are useful in various domains for example healthcare, smart city, transportation, and communication, etc. we presented various studies in these domains. A lot of researchers devoted their research in this direction. For instance, Minopoulos et al. (2022) presented the role of emerging technologies in the healthcare domain. They highlighted the role of modern IoT, Big data, and cloud computing in the detection of viruses, disorders, and illnesses. According to the researchers, the integration of mixed reality (MR) with Tactile Internet (TI) is helpful for the medical staff and also helpful in providing more accurate diagnosis and treatment of serious patients. At first, the authors highlighted the role of big data analytics especially in the healthcare domain. They proposed a system architecture that combines various future technologies. Similarly, Sahalu et al. (2022) presented recent advancements in emerging technologies in healthcare. In this study, At first, they highlighted the role of artificial intelligence (AI), blockchain, and IoT in providing healthcare services. They proposed a classification catalog and also presented various studies along with pros and cons. Finally, they discussed many ways for large-scale data handling, security, and privacy aspects in AI. Another interesting research carried out by Puspitasari et al. (2023) explored various machine learning-based emerging technologies for 5G communication networks. In this research study, at first, they highlighted unmanned aerial vehicles (UAVs) and intelligent reflecting surfaces (IRS) as modern communication technologies used for reducing overhead, massive users, and computational complexity. They could solve the concerns of 6G requirements. According to them, the incorporation of machine learning (ML) is used to solve mathematical problems quickly. In addition, the incorporation of reinforcement learning (RL) with modern communication technologies could be a feasible solution. They explored various ML and RL-based models for modern communication systems. They discussed the 6G wireless networks and also suggested ML solutions for future researchers. Sultan, Ali & Zhang (2018) presented a big data perspective for the next network generation networks, i.e., 5G communication networks. In terms of big data the achievement of a high data rate, low latency and are the goals. They require architecture design, and also upgraded infrastructure. In this survey, the authors identify many opportunities for the implementation of 5G networks. They have also presented many solutions related to energy conservation for these networks. Quesada et al. (2019) presented research in the integration of complex networks and ML. According to the authors, the CNA-based methods combined with ML open new ways in the modern world. After careful investigation, we concluded that most of the authors focused on the integration of limited emerging technologies except SIoT with communications networks, etc. None of them suggested the with respect to the integration of complex networks and SIoT. Thus in this subsection, we first explain the complex networks along with a basic definition and then recent research in this era. In ‘Leading Emerging Technologies and Recent Research’, we discuss the integration of complex networks with the leading emerging technologies.

Figure 2 Complex networks and emerging technologies.

Illustration credits: Iot (FreePik, https://www.flaticon.com/free-icon/iot_6702322?term=internet+of+things&page=1&position=4&origin=search&related_id=6702322); SIoT (Becris, https://www.flaticon.com/free-icon/networking_1239682?term=network&page=1&position=15&origin=search&related_id=1239682); Big data (Aficons studio, https://www.flaticon.com/free-icon/big-data_5206221?term=big+data&page=1&position=6&origin=search&related_id=5206221). Flaticon License.

Complex networks

In this sub-section, we discuss complex networks and the recent research including basic definitions and key concepts of complex networks. Later, we explain scale-free, random, and small-world networks within connection to social networks. A complex network has a non-trivial structure. The study of complex networks involves analyzing the patterns and properties of the network. Complex network theory is a branch of mathematics and computer science that deals with the study of complex networks (Fan et al., 2020). Table 1 presents the major classes of complex networks along with applications. In this table, readers can see that social networks are comprised of a group of people either friends or families. The technological networks comprised of internet and transportation networks. It is concluded, that major classes have common properties. Many researchers are devoted to research in this direction for instance; Shishavan & Gharehchopogh (2022) proposed a search optimization model using a genetic algorithm (GA) (Lyu, Wang & Zhang, 2020) for the detection of a community in complex networks. According to the authors, GA in community detection is helpful to increase accuracy and speed. Ghafori & Gharehchopogh (2022) presented a multi-objective search algorithm for community detection. Gharehchopogh & Abdollahzadeh (2023) presented a review of the leading search algorithms. These algorithms are categorized based on robustness and efficiency. A survey was conducted by Gharehchopogh (2023b). The authors collected a database based on the quantum metaheuristic (Hakemi et al., 2022) and proposed an algorithm for route optimization. Gharehchopogh et al. (2023) performed another survey on meta-heuristic. He found that these algorithms provide novel solutions and are also applicable to optimization problems. Another research was conducted by Gharehchopogh (2023a). He presented another improved optimization algorithm for the detection of a community in social networks. The main objective is to identify the structural properties of complex networks.

Table 1 Major classes of complex networks.

Technological Networks	Internet, Transportation, Communication Systems	
Ecological or biological networks	Disease, Epidemics, Protein networks, Neural system	
Economic networks	Corporate, Partnerships, and Financial Transactions	
Social networks	Professional Groups, Friends, and Families	
Cultural networks	Art, History and literature, Semantic networks, Religion networks	

Network, graph, and metrics

A network is symbolized by using a graph (Sohn, 2016). Graphs are used in computer science and mathematics disciplines. These are used to model and analyze complex systems. The sensors, computer networks, brain networks, and social networks can be denoted as a graph. Generally, a graph is a collection of nodes and edges that connect pairs of vertices. These are commonly used to study the relationships between objects, such as social networks and road networks, etc. A graph is a collection of vertices connected using edges. In particular, each edge joins exactly two vertices. A graph comprised V vertices and E edges, G = V, E.

Path length

In a graph, the path length is the distance between one vertex to another (Amin, Barukab & Choi, 2023). The average path length is used to assess the size of the network. It is measured by adding all the minimum distances between i and j. (1) l=N2−1 ∑i≠jLij

where N denotes the total number of vertices. The distance is denoted by node I and, j by, and N choose 2 denotes all possible numbers of pair vertices.

Clustering coefficient

It is a measure of the degree to which the graph exhibits local clustering. This indicates the cliquishness of a typical friendship circle. This property is calculated by adding the actual number of links connecting the neighbors of node i and j. (2) C=1N∑i=1NCi

where Ci is the clustering coefficient for vertex i. The average clustering coefficient (CC) of overall vertices i = 1……..N is given below. (3) Ci=2Eikiki−1

where Ei denotes the actual number of edges connecting the neighbors of vertex I, ki is the total number of neighbor vertices connected to vertex I and the maximum possible connections between neighbor vertices are kiki−1.

Small world graph

A small world graph is a type of network where the nodes are not directly connected and can reached through a small number of connections. This model initially starts with a ring having Nnodes and all nodes and the links are rewired together with the probability of p. In addition, the nodes I and j are chosen randomly. The small world phenomenon (Newman, 2000) refers to the principle that all people are connected. The basic idea of six-degree separation is that people are connected using a short chain and it has at least six people long.

Classical random networks

A classical random network (Newman, 2018) is a type of network where nodes are connected at random and without any particular pattern. This ring network is constructed using N nodes. The nodes I and j are connected by probability p. This process is repeated for all the possible node pairs, N(N − 1)/2.

Scale-free and growing graphs

A network whose degree distribution follows a power law. One of the common features of real-world networks is the presence of hubs in the network. These hubs were connected to the network. Scale-free network networks were characterized by the presence of large hubs. The degree distribution of a scale-free network exhibits the power law property with a power exponent of α > 1. One of the key examples of real-world networks with power law distributions are protein networks and e-mail networks. A comparison of complex network models along with network metrics is shown in Table 2. In this table, a reader can see that the scale-free model embodies power law and performs well compared to other models in case of failure. In addition, it has a short path length. The small world and classical random networks have poison distributions and are worst in case of link failure.

Propose methodology

In this section, we present our proposed methodology. Herein, we explain how we collect and prepare the data for our RQs. To understand modern technologies, we need to take an approach to classifying these technologies. To ensure comprehensive coverage of the literature on this topic, a multi-faceted approach is crucial. The details of the search methodology are given below.

Database and journal search

We have used various academic databases such as Google Scholar, IEEE Xplore, Science Direct, etc. These databases are used to gather a broad range of journal research articles and conference papers. This includes specialized journals in the fields of engineering, and information technology.

Keywords and search criteria

At first, a broad search was conducted to classify the IoT domains. We conducted a depth-wise search to investigate state-of-the-art research in these areas. Furthermore, based on our formulated research questions, we applied composite search strings to study the research topics. For this, we developed a comprehensive list of keywords and phrases such as “emerging technologies”, “IoT”, “SIoT”, “Big data”, and “social networks”. We suggested Boolean operators to expand or narrow the search results (e.g., “IoT AND ‘Emerging technologies” or “smart city OR ‘sustainable living”’).

Table 2 The comparison of complex network models and network metrics.

	Complex networks	Models	Degree distribution	Robustness	Path length	Clustering coefficient	
1	Scale-free network	Barabási and Albert (BA)	Power law	Strong	Short	High	
2	Small world graphs	Watts and Strogatz (WS)	Normal/ Poisson	Worst	Short	High	
3	Classical random network	Erdös and Rényi (ER)	Normal/ Poisson	Strong	Short	Low	

Inclusion and exclusion of grey literature

Herein, we have incorporated grey literature such as technical reports, government publications, and white papers to ensure coverage of non-commercial, cutting-edge innovations and regulations in this area.

Leading Emerging Technologies and Recent Research

Internet of Things

IoT is a trending and leading technology in the modern era. In IoT, the physical objects, sensors, and actuators, were connected to the internet. Figure 3 presents the conceptual view of IoT where various devices from the physical, virtual, and digital worlds are connected. These devices can send and receive data from the IoT cloud server. For communication, different communication technologies are used and known as gateways. The data is forwarded using a transit network named IPv4. The key responsibility of the IoT cloud server is to provide the processing, storage, and data analysis. The user view provides various applications for instance; smart factory, smart health care, smarty transportation, etc. This technology is used in a variety of domains and moving out of traditional information and communications technology (ICT) (Ghasempour, 2019). Figure 4 illustrates the technological roadmap of IoT in the last two decades. The technological change begins with radio-frequency identification (RFID) devices and moves towards the ability to attach distinct devices. In addition, this brings security and mobility features. A lot of researchers work in this direction. For example, Ghasempour (2019) surveyed various IoT applications using a smart grid. The smart grid is one of the most important applications of IoT and is used to solve the issues of the electricity grid. The authors presented a relationship between the smart grid and IoT. Similarly, Saha, Mandal & Sinha (2017) presented research on the most recent trends in the IoT. In this research, they first presented an overview of different recent trends in IoT. Later, highlighted the most important applications in autonomous control, cloud computing, and AI. Similarly, Shiddike et al. (2022) proposed a smart grid management system using blockchain for the IoT. The security is guaranteed by the on-demand and authorized users using blockchain technology. The data is stored on the blockchain server. Orlando et al. (2022) proposed a smart meter infrastructure for smart grid IoT applications. In this research, they proposed multiple algorithms for smart grid management. These algorithms could add, update, and delete the data. One of the key features is ‘cost’. The cost is reduced and improves the scalability. Lõpez Peña & Fernández (2019) proposed an architectural model for achieving high performance. A paradigm is presented using cloud computing for achieving transparency. Singh et al. (2020) present a review of blockchain and artificial intelligence. In this research, they discussed various challenges. Liu et al. (2023) proposed an efficient framework combining the cloud and AI for making efficient analyses of big data. Table 3 presents the summary of the most recent advancements in different application areas of IoT along with the pros and cons.

Figure 3 The conceptual view of IoT.

Illustration credits: Cloud (iconixar, https://www.flaticon.com/free-icon/cloud_1163624?term=cloud&page=1&position=5&origin=search&related_id=1163624); Datavase (phatplus, https://www.flaticon.com/free-icon/database-storage_2906274?term=server&page=1&position=8&origin=search&related_id=2906274); Switch (Ranah Pixel Studio, https://www.flaticon.com/free-icon/network-switch_9655733?term=switch+router&page=1&position=11&origin=search&related_id=9655733); swich 2 (Hopstarter, https://www.flaticon.com/free-icon/internet_11892161?term=switch+router&page=1&position=41&origin=search&related_id=11892161); switch 3 (srip, https://www.flaticon.com/free-icon/wifi_1976620?term=switch+router&page=1&position=79&origin=search&related_id=1976620); Sensor 1 (Iconjam, https://www.flaticon.com/free-icon/motion-sensor_9708985?term=sensor&page=1&position=9&origin=search&related_id=9708985); sensor 2 (gravisio, https://www.flaticon.com/free-icon/wearable-technology_12350616?term=sensor&page=1&position=3&origin=search&related_id=12350616 sensor 3 https://www.flaticon.com/free-icon/temperature_3593387?term=iot+sensors&page=1&position=5&origin=search&related_id=3593387.

Figure 4 Technology roadmap of IoT.

Social Internet of Things

SIoT works as a cluster between IoT and social networks (Khan et al., 2021). This technology is developed to overcome the practical and key challenges of IoT, i.e., scalability, massive object-to-object interconnection, object discovery, etc. The SIoT refers to the convergence of IoT and social networks. It enables developers to create a network in which the object utilizes the power of social relationships. In SIoT the object interacts with others and also behaves socially. The object can request and also provide required services. Table 3 presents the overview of the most recent advancements and developments in IoT. The SIoT was coined in 2013. One good example of SIoT is autonomous vehicles. In these vehicles, the embedded sensors can communicate with each other to avoid road accidents. As the SIoT is a new and emerging technology. It has no standard framework and simulator available for simulation. A lot of research carried out in this direction, For instance, Mohana, Prakash & Krinkin (2023) presented a simulator for the SIoT using AI named CCNSim which is used to overcome the buffer overflow issues in IoT. The proposed model is analyzed and tested for IoT applications. Similarly, Amin, Ahmad & Choi (2018) explored various simulation tools, i.e., Igraph, Pajek, Netminer, and Gephi for social network analyses. Randjelović & Popović (2011) presented various social network analytics tools for criminology: Igraph, Pajek, etc. Amin et al. (2022) presented a systematic survey on the recent advancements in SIoT. In this survey, they have classified the SIoT into six areas network navigability, service composition and discovery, architecture components, relationships and trust management, and architecture and components, etc. They have presented an overview of SIoT using big data, IoT, and computing. They identified various challenges, for example, security, data management, and energy management. Fadda et al. (2023) presented a solution for the monitoring of traffic and pollution in a smart city. The proposed system analyzes and tracks the vehicles moving on the road and in pedestrian areas. It can monitor and analyze the air quality by using specialized sensors. The proposed solution is equipped with the appropriate intelligence and taken into account for instance speed traffic and pollution.

Table 3 Overview of most recent advancements and developments in IoT.

Reference	Description	Application area	Advantages	Disadvantages	
		Smart Grid	Could Computing	Artificial Intelligence	Block Chain			
 Ghasempour (2019)	A framework is presented using blockchain.	✓	×	×	✓	Security is provided.	It is not tested in a controlled environment.	
 Saha, Mandal & Sinha (2017)	A distributed three-phase software infrastructure is proposed using metering technology.	✓	×	×	×	The cost is reduced and scalability is increased.	It is not implemented for embedded systems.	
 Shiddike et al. (2022)	A model is proposed for achieving high performance using cloud computing.	×	✓	×	×	High performance.	It takes a lot of time to process.	
 Lõpez Peña & Fernández (2019)	A model is presented for marketing using AI and cloud computing.	×	✓	✓	×	It has a quick response time, a request rate & the ability to process a large amount of bank data.	Processing time is slow.	

Big data

Big data refers to extremely large data. The traditional data processing techniques and software are unable to handle large-scale data. The big data are typically characterized by three primary features, known as the Vs of big data. Volume refers to large-scale data in terabytes or exabytes. Secondly, velocity refers to the speed at which the data is generated, collected, and processed. Third, the variety refers to the diverse types of data, including structured data, semi-structured data, and unstructured data. Big data analytics is used in various domains such as; healthcare, finance, and business. The ultimate goal of big data analysis is to analyze the useful information and make intelligent decisions. In this direction, Yang & Yang (2017) explored various tools, for example Gephi, Citespace, and SCI 2. These tools are used to analyze and the processing of data. In addition, deep compassion is performed based on data processing, network extraction, and visualization.

In Table 4, a comparison of big data tools that support complex networks is presented. The comparison is performed based on data processing, network extraction, and visualization. In this table, the reader examines how the exact big data analytics tool has different features for the processing, extraction, and analysis of data.

Table 4 Comparison of big data tools support complex networks.

Comparison	Citespace	SCI2	Gephi	
Data processing	Duplicates removal			×	
Add merge synonyms			×	
Network extraction	Data selection		It is possible for a temporal period.	It is possible for a temporal period.	
Algorithm	Pathfinder and Minimum Spanning Tree (MST).	Pathfinder and Minimum Spanning Tree (MST).	×	
Visualization	Expression of information	It is available by frequency, time, and centrality.	It is available by either frequency or centrality.	It is available by either frequency or centrality.	
Edit	This feature is the worst.	This feature is flexible.	This feature is flexible.	
Layout	Cluster view, Hybrid network, Timezone, Burst detection.	It is shown in various co-occurrence maps/Burst detection/Timezone.	It is shown in various co-occurrence maps/cluster views.	

The Synergies between Complex Networks and the Leading Emerging Technologies

Figure 5 shows the synergies between complex networks and leading emerging technologies. This figure summarizes the common key research areas of complex networks and the leading emerging technologies. The complex network can be modeled as scale-free, random networks, and other networks, etc. Similarly, scale-free networks are used in social networks and also in SIoT. The famous algorithms for link selection in complex networks are usually used in SIoT to achieve higher network navigation. In addition, complex networks are used for big data analysis. Big data methods are used for the data storage. The indexing and the processing of data have been performed. The research studies have shown a comparative analysis of recent developments in complex networks and internet-based technologies. For example, Chung & Sohn (2023) conducted a comprehensive review of complex networks using neural networks. This research is based on the key properties of neural networks. They covered small-world, scale-free, and random models. Another research carried out by Amin et al. (2022) explored next-generation advances in IoT. They partially covered IoT and complex networks are not covered. Similarly, Amin & Hwang (2021) presented a comprehensive review of network science and data science. The authors partially covered big data and did not cover complex networks. Wu et al. (2021) conceptualize the IoT as complex networks. They explained the concept of complex networks and the key properties of IoT. In addition, also presented a detailed multilayer IoT architecture. But did not explore big data and SIoT. Mata (2020) presented a mini-review on complex networks and explained fundamental quantities. However, they did not explore big data, SIoT, and the IoT. Amin, Ahmad & Choi (2018) partially covered various tools in complex networks. These are used for mining and community detection (Amin, Choi & Choi, 2020). They partially cover SIoT and do not cover the IoT and big data. Khan & Niazi (2018), presented emerging topics in IoT technology using complex networks. They cover complex networks but partially cover IoT. Yang & Yang (2017) presented a comparison of leading complex network analysis (CNA) software, i.e., SCI2, Gephi, Citespace, etc. This research is interesting and timely; however, they did not explore leading emerging technologies. Shah & Shah (2015) proposed a survey on the performance enhancement attributes in complex networks. They partially cover big data, IoT, and SIoT. Karimi et al. (2014) explore the recent advances and the developments in modern complex networks. Table 5 presents a summary and a comparison of our research and the most recent surveys. A reader can see that our survey completely covers complex networks and internet-based technologies but, others partially or have not covered. Complex networks are also useful in the development of smart cities and grids using IoT. Currently, there is no standard approach available for the modeling of real-world IoT networks. Therefore, to solve this issue, Batool & Niazi (2017) propose a standard model using complex networks and agent-based modeling. The proposed model uses Cognitive Agent-Based Computing (CABC) to simulate complex IoT networks. The authors demonstrate the modeling of several standard complex network models such as random, lattice, small-world, and scale-free networks. Sohn (2017) suggested small-world and scale-free networks. These models are used to reduce the network and device complexity. According to the authors, it can be applied to the IoT network to enhance error tolerance and synchronization. The proposed models are evaluated using various complex network metrics. Bonato & Tian (2012) surveyed and gave an overview of complex networks and social networks. In this research, they explained small-world networks in contrast to social networks. Amin et al. (2019) proposed a model for service search in a social network using SIoT. According to the authors, due to massive object interconnection, scalability arises in the IoT. To overcome this issue the incorporation of social networking concept into the IoT is a possible solution. They proposed certain rules for the next-hop neighbor search. These are derived from small-world networks. Based on these rules the next hop search is performed in a social network. The achieved results are efficient, however, restricting the number of users per node is a challenging task. Another research carried out by Amin & Choi (2021) proposed a service search model for achieving higher network navigation. The service search is carried out immediately once the service seeker initiates the service request. The service search is performed based on ‘centrality’. The problem with this model is that it can’t guarantee trust between the objects. The complex networks are used in state space and also for the prediction of random walkers for the next visit to a social network (An, O’Malley & Rockmore, 2019). For this, An, O’Malley & Rockmore (2019) proposed a model for the next hop prediction of a walker. They suggest network centrality for a next-hop walk. Sun, Medo & Staab (2020) research on time-invariant degree growth. They suggest a preferential attachment (PA) network. The PA is combined with fitness. Lacasa et al. (2008) presented a fast computational model. This model converts the time series into graphs. Zou et al. (2015) investigated the issue of event-triggered state estimation using complex networks. For this, they used the Lyapunov theory combined with the stochastic analysis (Zou et al., 2015). Generally, the IoT and SIoT generate large amounts of data, which can be analyzed using big data analytics to extract valuable insights. In this direction, Flouris et al. (2017) proposed a classification of complex event processing. They suggest the integration of complex networks and big data. Stefanowski, Krawiec & Wrembel (2017) explored big data and complex networks. In this research, the authors explored various solutions for the processing and storing of data. Figure 6 explains the holistic view of leading emerging technologies. We can see that end devices for example; cameras, smart traffic light sensors, and smart home devices were connected using the cloud (Marjani et al., 2017). These smart devices can carry a large amount of data called big data (Dong, 2022). The data has different formats. Initially, the data is stored in storage media using cloud technology (Deng et al., 2021). This data has been gathered using big data analytics. The big data operations were performed by using 5 Vs and the data complexity is reduced. Now the incoming data is stored in a large storage space. The last step is to perform an analysis of this data using big data tools for example; Spark (Fu, Sun & Wang, 2016), MapReduce (Siledar, Deogaonkar & Pagare, 2021), etc. In general, the problem occurred during data collection. This issue is solved by using IoT technology. The intelligent data pre-processing techniques are named pre-processing and meta-data creation. The IoT end devices forward data to different locations and are then processed. As the data size is very large; therefore, it is sent for processing from different locations. Assahli, Berrada & Chenouni (2017) presented a comparative study on the cleaning and preprocessing of IoT data. To do that, they suggested multi-agents. These agents are employed to reduce the complexity. It is evident from the literature that complex networks are also helpful in providing security for IoT devices. Ahmad et al. (2023) presented a complex network-based approach for providing security for vulnerable IoT devices. They proposed a framework for the identification of influential spreaders. These influences play a key role in providing the security feature. IoT devices are protected from malware attacks (Khan et al., 2024). The connectivity between the IoT devices is considered a network (Kumaran & Sridhar, 2020). The centrality measures are considered as the multi-attribute. In this way, security is provided using centrality measures. The problem with this research is that they did not experiment in a real environment. Table 6 presents a summary of recent research in complex networks and three aspects.

Figure 5 The synergy of complex networks and leading emerging technologies.

Illustration credits:IoT (Freepik, https://www.flaticon.com/free-icon/iot_6702322?term=internet+of+things&page=1&position=4&origin=search&related_id=6702322); SIoT https://www.flaticon.com/free-icon/people_12820596?term=people+link&page=1&position=44&origin=search&related_id=12820596 Big data (Aficons studio, https://www.flaticon.com/free-icon/big-data_5206221?term=big+data&page=1&position=6&origin=search&related_id=5206221); Hand https://www.flaticon.com/free-icon/point_556130?term=hand&page=1&position=32&origin=search&related_id=556130 Flaticon license.

Table 5 Summary and comparison of recent surveys with the proposed survey.

References	Year	Complex Networks	Internet of things	Social Internet of Things	Big Data	
This research	2024	✓	✓	✓	✓	
Chung & Sohn (2023)	2024	✓	×	×	×	
Amin et al. (2022)	2023	×	✓	×	✓	
Amin & Hwang (2021)	2021	×	×	×	∘	
Wu et al. (2021)	2021	∘	✓	×	∘	
Mata (2020)	2020	✓	×	×	×	
Amin, Ahmad & Choi (2018)	2018	∘	✓	×	×	
Khan & Niazi (2018)	2018	✓	×	×	×	
Yang & Yang (2017)	2017	✓	×	×	×	
Shah & Shah (2015)	2015	×	×	×	×	
Karimi et al. (2014)	2014	∘	×	×	×	
Notes.

✓ Covered

× Not Covered

∘ Partially covered

Figure 6 A holistic view of leading emerging technologies.

Illustration credits: Traffic (xnimrodx, https://www.flaticon.com/free-icon/traffic-light_3465353?term=traffic&page=1&position=13&origin=search&related_id=3465353); Home (freepik, https://www.flaticon.com/free-icon/house_619032?term=home&page=1&position=16&origin=search&related_id=619032); www (NajmunNahar, https://www.flaticon.com/free-icon/www_10254207?term=www&page=1&position=33&origin=search&related_id=10254207); windmill (Freepik, https://www.flaticon.com/free-icon/windmill_1926396?term=windmill&page=1&position=1&origin=search&related_id=1926396); Camera (dreamicons, https://www.flaticon.com/free-icon/cctv-camera_2642651?term=camera&page=1&position=17&origin=search&related_id=2642651). Flaticon License.

Table 6 Summary of research in complex networks and leading emerging technologies.

References	Contribution	Key Technology	Applications	
Batool & Niazi (2017)	✓Develop a standard for the IoT using complex networks.
✓ The authors used agent-based modeling.	Agents and complex networks	✓ Measure energy consumption by nodes.	
Sohn (2017)	✓Develop a model for IoT systems using small-world and scale-free networks.	Complex networks	✓ Develop a hybrid complex network and IoT model for node energy consumption.	
Amin et al. (2019)	✓Develop a model for the service search/link selection using SIoT.
✓ The key properties of small-world networks have been used in the model.	SIoT	✓ Service search and smart city.	
Ahmad et al. (2023)	✓Developed a complex network-based approach for providing security to highly vulnerable IoT devices in a smart city.
✓ The security is provided using centrality measures.	Complex networks	✓ Security.	
Zhang et al. (2022)	✓Developed an approach for the evaluation of node importance in complex networks using deep learning.	Deep learning	✓ Network robustness is granted.	

Discussion, Open Research Challenges, Potential Solutions, and Future Research Trends

Discussion

The discussion around IoT is vast. Herein, in this paper, we cover both its potential in industries and the hurdles that must be overcome to achieve worldwide implementation. We have divided this part into subsections.

Opportunities for growth

IoT is developed to develop and propose new applications and also accelerate new research areas such as smart cities, smart healthcare, and smart manufacturing. For example, smart cities implementing IoT can improve traffic management through smart sensors that monitor road congestion and also adjust traffic signals (in real-time). Recently, it has been used in the industrial sector. This recent development is named as the Industrial Internet of Things (IIoT). IIoT is seeing a massive productivity gain, where companies develop new technologies and these technologies can used to reduce machine downtime through predictive analytics.

Privacy and ethical concerns

One of the key concerns in IoT is privacy. As mentioned earlier, many IoT devices collect the data. For example; whether it’s through fitness trackers, home security systems, or smart cars. Thus, it raises significant privacy concerns. As more personal data becomes accessible, the risks of surveillance, misuse, or unauthorized access highly increase. In addition, the ethical considerations around data ownership and consent have become a critical part of the IoT, especially as it interfaces with AI.

The future of IoT: standardization and regulation

As IoT reaches its full potential. Thus, the issues related to standardization and regulation must be addressed carefully. In this regard, the governments and industry leaders propose unified protocols. These will allow devices from different manufacturers to communicate seamlessly. Additionally, regulatory frameworks will need to be developed to govern data privacy, cybersecurity, and liability in cases of IoT device failures.

Security risks and mitigation

The security remains one of the most significant challenges in IoT. As a large number of devices are connected, this creates a larger surface area for potential cyberattacks; in this scenario, the cyber attacks, for example Mirai botnet, which took over hundreds of thousands of IoT devices. In 2016, underscored the need for better security protocols. The industry is focusing on embedding security at the hardware level and using encryption, authentication, and regular software updates to mitigate these risks.

IoT’s role in sustainability

IoT plays a pivotal role in promoting sustainability by optimizing resource use and energy conservation. For example, smart grids can regulate energy usage in homes and businesses based on real-time data, reducing wastage. Similarly, in agriculture research, IoT-driven precision farming techniques use sensors to monitor the soil conditions and also used to optimize irrigation. Finally, reducing water usage and also increasing crop yields.

Open research challenges

The open research challenges in this area are given below.

Complexity

The traditional IoT is a large-scale heterogeneous network. It has a complex system without central control. Thus, it has structural complications. We named as complexity. In addition, other challenges include the interaction of complex factors and dynamic nodes. These all factors reflect the resilience and robust nature of IoT.

Connectivity

IoT supports large object connectivity around the globe. The key challenging task is to connect everything (devices) to the internet and provide reliable and fast network connectivity.

IoT standard platform

Currently, there is no standard platform available for the development of IoT. Thus, there is a need a propose an IoT standard platform for developing applications.

Complex data

In IoT, the data is aggregated from multiple attached devices and they exhibit heavy tail behaviors (because of no trivial dependence on IoT). In addition, the data is heterogeneous and has different types for example; videos, audio, images, and text, etc. The diverse data needs to be cleaned, transformed, and processed. The combining of unstructured data and reconciling them for use in report creation is incredibly difficult to achieve in real-time. Therefore, the data complexity issue arises as a common problem in IoT.

Noisy data

The data is collected from various large-scale networks. Thus it consists of diverse types of data and it has errors and missing values. It has the possibility of noise. Secondly, when the data is heterogeneous. This refers to the ‘variety’ of big data. The challenge is to clean the large amount of data.

Big data collection and storage

In general, a large data is passed through IoT devices. If the developer has good knowledge, he decides the format and the data collected from the sensors. If he doesn’t have it may lead to the collection of garbage data and hence the cost will be increased. Herein, the problem of data collection, processing and analysis, and visualization arises. The issue of 5 Vs arises especially when dealing with IoT technologies. We need advanced data analytics and machine learning tools to extract meaningful insights from data.

Self-organized structures

The IoT devices communicate with others and provide various services. For instance, autonomous vehicles, smart road systems, and smart planes are used in smart cities. Advanced and novel IoT solutions can be developed and extended by learning systems and automated reasoning. Without applying sufficient information to human interaction and decision support. Further efforts have been focused on the development of novel IoT solutions that are capable of deciding, interacting, learning, and perceiving.

Social IoT observations

In SIoT the connections are established between the devices based on social relationships in a similar way as human relations. This technology enables the devices to achieve higher network navigability. The network models that are used to study human behavior are similar to the study of social networks in IoT devices. Currently, SIoT lacks architectural models and simulation tools.

Trust, security, and privacy

Since IoT devices collect a large amount of sensitive data, they are prone to cyberattacks. Ensuring the security at both the device and network level is a crucial challenge. Data breaches can lead to severe privacy violations, especially when dealing with personal data, like health information. Trust is a main concern in the development of IoT-based applications. It is directly related to privacy and security. As the devices and the people are involved in IoT applications, trust arises as a major issue. During data transmission, it is necessary to secure the data. When these devices are connected to the internet there is a probability of attacks on these devices. Similarly, in the case of SIoT, the relationships were established between nodes based on friendship and trust level. In this scenario, the selection of dishonest nodes in the network will affect the overall performance of the network. In addition, the storing of a large history of transactions on objects will lead to scalability and also in terms of communication overhead. Sometimes the mathematical calculation in these objects may generate computation overhead. Thus, ensuring trust between objects in IoT-based technologies is a very serious and challenging task.

Energy management

As IoT is comprised of tiny devices and these are connected via the internet. During the data transmission process, the energy consumption in these devices arises. Therefore, energy consumption is a serious issue in IoT.

Scalability

IoT networks can include millions of devices, making scalability a major issue. Network congestion, data storage, and processing become critical as IoT implementations grow in size and complexity.

Potential solutions

It is evident from recent literature, that complex networks and the network theory are used for the development of novel experimental and theoretical frameworks. In addition, these are also helpful in constructing and extracting meaningful information using theory measurements such as entropy and network functions. From this aspect, we can say that complex networks can be used as a solution for the IoT. Similarly, edge computing and cloud computing can be combined to develop and propose novel applications for IoT. The combination of blockchain and software-defined networks (SDN) are leading technologies to ensure network connectivity and secure systems. In addition, these technologies are used for energy management in IoT-based technologies. The SIoT comprises users holding smart devices. During device operation, a large amount of energy is consumed during coordination. In this scenario, energy consumption is a coordination factor in the operation and the design of SIoT. To overcome this issue a new term has been introduced in recent days named Social Internet of Energy (SIoE). This technology allows the physical devices that consume more energy to create social relationships and hence the overall scalability is improved. The SIoE is used to develop novel applications for smart grids. The smart grid is widely used in smart cities.

Future research trends

Research into IoT has exploded in recent years. The key emerging research areas include:

Artificial intelligence and machine learning: a powerful synergy

These technologies are increasingly integrated into IoT systems to enable more sophisticated decision-making. AI can help predict patterns in data and automate responses in real-time.

Analyze massive datasets

As IoT generates enormous volumes of data from sensors, devices, and machines. Thus, AI and ML provide the tools to process, analyze, and interpret this data in real-time.

Predict outcomes

By leveraging historical data and learning from it, ML algorithms can predict future events, such as equipment failure, traffic congestion, or energy usage patterns.

Automate decision-making

AI systems enable IoT devices to take autonomous actions based on insights derived from the data. This can include automating industrial processes, adjusting environmental controls in smart buildings, or optimizing logistics routes.

Federated learning

Federated learning (FL) is a leading technology. In FL, the ML models are trained across multiple devices without sharing raw data. This is particularly valuable for IoT applications where privacy is paramount, such as in healthcare. The data stays on local devices, and only model updates are shared, mitigating privacy risks.

AIoT (Artificial Intelligence of Things)

Artificial Intelligence of Things (AIoT) presents the combination of AI with IoT. The key objective is to create more intelligent and autonomous systems. This will enable applications like smart cities where traffic control, waste management, and energy distribution are autonomously optimized by AI algorithms.

Blockchain technology

Blockchain is being explored as a solution to address security and privacy challenges in IoT. The blockchain is used in decentralized ledgers to ensure data security and data sharing between different devices.

5G networks

The emergence of 5G is seen as a new catalyst for IoT growth. 5G’s technology has low latency, high bandwidth, and faster data speeds. These all enable more devices to be connected and operate in real-time. It is used in fostering applications for example; autonomous vehicles, etc.

Sustainability

IoT is also being leveraged for sustainability initiatives, such as optimizing energy consumption and reducing emissions in smart cities.

Complex networks

Developing multilayer networks reduces the complexity in terms of cost and energy. Thus, future exploration of network complexity in IoT will develop the network structure and the functions of IoT.

Emerging technologies

The development of novel solutions empowered by ML and AI is necessary. IoT software and hardware developers might have to explore complex networks for the development of new standards. It is evident from the research that complex network tools are used for the IoT and SIoT, for example, Pajek, Netminer, and Igraph, etc. To solve this issue, there is a need to develop advanced AI and ML-based solutions with the combination of complex networks. The researchers should focus on the development of novel and advanced models for the removal of noise from data. The development of novel solutions in this area that can draw a picture of huge amounts of data, and provide actionable insights is important for the researcher’s community. The realization of this vision, self-organization, and autonomous IoT infrastructure will enable the ongoing research in the area of semantics for interpretability and intelligent cognition and learning big data. Increasing the computation power has enabled complex network tools and mathematical simulations at all scales in social IoT to reach more investigators. In this research direction, research questions are as follows: What entities are available in the IoT-based technologies? How this can make the trust management process more efficient and accurate? Any efficient trust models are available in the IoT? What models and tools are available for security and privacy? To answer these questions new models should be developed. The key objective of SIoE is to enhance the relationships established between the devices and users in SIoE. But, the integration of SIoE and SIoT is still a challenging task. In this context, new models for efficient energy consumption are a new area in future research. Table S1 explains the challenges, contributions, future research directions, and remedies.

Conclusions and Future Works

IoT technology has rapidly emerged as a revolutionary field in computer science, facilitating the connection of everyday objects to the internet and enabling a vast network of interconnected devices. In this research, we carried out a literature survey on the next generation Internet of Things using emerging technologies. First, a review of complex networks, basics, and definitions is presented. We discuss various complex network models including scale-free, random, and world networks, etc. Then we presented various useful complex network metrics for social networks, for instance; path length and the clustering coefficient. This survey compared the most recent state-of-the-art in this area in since last 10 years, i.e., IoT by definition and from a conceptual view is well explained. In addition, a road map for the IoT is presented. SIoT is a new technology well explained along with recent applications. Finally, we presented a detailed discussion on energy management, trust security and privacy, and big data collection. In the future, this will explore machine learning, artificial intelligence, and cloud computing in connection with complex networks.

Supplemental Information

Table S1 Summary of future research challenges and potential solutions

Additional Information and Declarations

Competing Interests

Author Contributions

Data Availability

The authors declare there are no competing interests.

Farhan Amin conceived and designed the experiments, analyzed the data, prepared figures and/or tables, authored or reviewed drafts of the article, and approved the final draft.

Rashid Abbasi conceived and designed the experiments, analyzed the data, authored or reviewed drafts of the article, and approved the final draft.

Salabat Khan conceived and designed the experiments, prepared figures and/or tables, authored or reviewed drafts of the article, and approved the final draft.

Muhammad Ali Abid conceived and designed the experiments, performed the experiments, performed the computation work, authored or reviewed drafts of the article, and approved the final draft.

Abdul Mateen conceived and designed the experiments, performed the computation work, prepared figures and/or tables, and approved the final draft.

Isabel de la Torre conceived and designed the experiments, performed the experiments, authored or reviewed drafts of the article, and approved the final draft.

Angel Kuc Castilla conceived and designed the experiments, prepared figures and/or tables, authored or reviewed drafts of the article, and approved the final draft.

Eduardo Garcia Villena conceived and designed the experiments, prepared figures and/or tables, authored or reviewed drafts of the article, and approved the final draft.

The following information was supplied regarding data availability:

This is a literature review.

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
