# Peer review of "Latest advancements and prospects in the next-generation of Internet of Things technologies"

_PeerJ Computer Science, doi:10.7717/peerj-cs.2434_

## Round 0.1 · original submission · Major Revisions

After carefully considering the reviews and assessing your manuscript, I am pleased to inform you that we would like to invite you to revise and resubmit your manuscript for further consideration. The reviewers have provided constructive comments that will help strengthen your work. Please address each of these points thoroughly in your revised manuscript. Additionally, ensure that you provide a detailed response letter outlining how you have addressed each comment raised by the reviewers. This will help the reviewers and myself to evaluate the changes made to the manuscript. It is PeerJ's policy that additional references suggested during the peer-review process should only be included if the authors agree that they are relevant and useful.

Reviewer 1 ·

Basic reporting

This research article offers a comprehensive literature review of the Internet of Things (IoT) and its associated emerging technologies. It addresses the challenges faced by IoT in terms of efficiency, architecture, complexity, and topology, highlighting the limitations of traditional technologies in these areas. The study delves into the role of complex networks in analyzing and improving IoT structures, introducing the Social Internet of Things (SIoT) as a promising advancement to tackle existing challenges. Additionally, it examines big data concepts, including the 5Vs, and various graph models related to social networks. Covering developments from the past decade, the article integrates discussions on complex networks, cyber-physical systems (CPS), and their synergy with IoT, SIoT, and big data. It concludes by outlining the impact of these emerging technologies on future applications and solutions, serving as a valuable reference for researchers and readers in the IoT domain.

However there are some important basic issiues in paper as mentioned;
1) Introduction is given in a very long paragraph. It repeats the message it wants to give, which becomes meaningless for the reader after a while.
2) Figure 1 in the introduction is not where it should be. In the peer-reviewed journal format, figures should be placed below the text. In this state, we cannot see or interpret Figure 1 when referenced. The same applies to all figures and tables in the article.
3) It is really good that they wrote 'Key Objectives and contribution' in separate title but they repeat themselfs. They have repeated in this section the things they already mentioned in introduction. Introduction should definitely be rewritten.
4)

Experimental design

Study design looks complicated for the reader. This is because the figures and tables are not in their proper positions. Also, there are too many headings.
1) Figures and tables must be in their place.
2) Formulas and their explanations are very successful.
3) The number of headings should be reduced, the article is too complex in its current form.
4) In 148 they said; '...various academic databases such as...' they should give all database names.

Validity of the findings

Since the article is a survey, it can be a successful research study in the field of IoT, but it has its own shortcomings.
1) Since there is no research question, we do not know exactly what they want to answer. They have put MC, but it is not clear for what purpose they put them.
2) There should definitely be a discussion section. In this section, what they aimed for with the survey and where they reached should be summarized.
3) There are no references to 2024. This is not possible since the subject of IoT is a very popular subject in computer science. The references should definitely be updated.

Reviewer 2 ·

Basic reporting

How is your work different? Add a novelty subsection
English is very weak..look at this line 180-181 "Figure 2 the connection between complex networks and emerging technologies
181 Figure 2 demonstrates the connection between complex network networks and emerging "
Proper citation style is not used. Look at the journal guidelines and citation style needs to be followed properly.
Whay have author kept Figure and Tbale captions in the manuscript. like Table 3 Overview of most recent advancements and developments in IoT... they need to be part of the tables/figures..
Read journal guidelines properly before submitting to this journal
Why authors have given so much of focus in the manuscript on basic terms like Big data, Social Internet of Things etc etc...not required this much

There is so much of theory in the manuscript..very hard to read.

Experimental design

Break the following "Discussion, Open Research Challenges, Potential Solutions, and Future Research Trends" as separate sections with at least 3 pages per section.It us very confusing . You can pack them all as one and just conclude. There should be a separate section on each in the revised manuscript as
Discussion
Open Research Challenges
Potential Solutions
Future Research Trends
What are the open research questions in this. Discuss them.

The manuscript is not in a position be accepted if all the issues are not addressed properly.
It needs a lot of work.
There are lots of authors in this manuscript and a good quality of work is expected after the revision.
I will see it carefully in revision if it is in position or not

Validity of the findings

What methodology was used to review? May discuss that

---

## Round 0.2 · Minor Revisions

Kindly address the missing final comments of one of the reviewers.

Reviewer 2 ·

Basic reporting

I had give a comment ": Break the following "Discussion, Open Research Challenges, Potential Solutions, and Future Research Trends" as separate sections with at least 3 pages per section. It us very confusing .
You can pack them all as one and just conclude. There should be a separate section on each in the revised manuscript as
Discussion
Open Research Challenges
Potential Solutions
Future Research Trends
What are the open research questions in this. Discuss them.
"

It is still not addressed.
Break the following as seperate sections and discuss them in details : "Discussion, Open Research Challenges, Potential Solutions, and Future Research Trends"

Whtaever you have mentioned in the revised manuscript is not acceptable. Revise it..

English needs improvements

Experimental design

no comment

Validity of the findings

no comment

---

## Round 0.3 · accepted · Accept

I am pleased to inform you that your paper has been accepted for publication in PeerJ Computer Science. Your manuscript has undergone rigorous peer review, and I am delighted to say that it has been met with praise from our reviewers and editorial team. Your research makes a significant contribution to the field, and we believe it will be of great interest to our readership. On behalf of the editorial board, I extend our warmest congratulations to you.